# Critical Evaluation of Sinonasal Disease in 64 Adults with Primary Ciliary Dyskinesia

**DOI:** 10.3390/jcm8050619

**Published:** 2019-05-07

**Authors:** Emilie Bequignon, Laurence Dupuy, Francoise Zerah-Lancner, Laurence Bassinet, Isabelle Honoré, Marie Legendre, Marie Devars du Mayne, Virginie Escabasse, Bruno Crestani, Bernard Maître, Estelle Escudier, André Coste, Jean-François Papon

**Affiliations:** 1AP-HP, Hôpital Henri-Mondor–A Chenevier et Hôpital intercommunal, Services d’ORL et de Pneumologie, Créteil 94010, France; emilie.bequignon@gmail.com (E.B.); laurencedupuy@hotmail.fr (L.D.); francoise.zerah@aphp.fr (F.Z.-L.); kilad@orange.fr (L.B.); mariedevars@gmail.com (M.D.d.M.); Virginie.Escabasse@chicreteil.fr (V.E.); bm.maitre@gmail.com (B.M.); andre.coste@chicreteil.fr (A.C.); 2INSERM, U955, Créteil 94010, France; 3Université Paris-Est, Faculté de Médecine, Créteil F-94010, France; 4CNRS, ERL 7000, Créteil 94010, France; 5AP-HP, Hôpital Henri-Mondor–A Chenevier, Service De Physiologie et d’explorations Fonctionnelles, Créteil 94010, France; 6Assistance Publique-Hôpitaux de Paris, Hôpital Cochin, Service de Pneumologie, Paris 75014, France; isabelle.honore2@aphp.fr; 7Université Paris Descartes, Sorbonne Paris Cité, Paris 75006 France; 8Service de Génétique et Embryologie Médicales, Hôpital Armand-Trousseau, Assistance Publique-Hôpitaux de Paris, Paris 75012, France; marie.legendre@aphp.fr (M.L.); estelle.escudier@aphp.fr (E.E.); 9INSERM, Unité Mixte de Recherche S933, Université Pierre et Marie Curie, Paris 75005, France; 10Maladies pulmonaires Rares (DHU FIRE) Service de Pneumologie A, Assistance Publique-Hôpitaux de Paris (AP-HP), Hôpital Bichat, Paris 75877, France; bruno.crestani@aphp.fr; 11Laboratoire d’Excellence Inflamex, Paris 75013, France; 12Université Paris Diderot, Sorbonne Paris Cité, Paris 75013, France; 13INSERM, Unité 1552, Paris 75013, France; 14AP-HP, Hôpital Kremlin-Bicetre, Service d’ORL et de Chirurgie Cervico-Faciale, Le Kremlin-Bicêtre 94275, France; 15Université Paris-Sud, Faculté de Médecine, Kremlin-Bicêtre F-94070, France

**Keywords:** airway mucociliary clearance, chronic rhinosinusitis, Kartagener syndrome, otitis media with effusion, primary ciliary dyskinesia

## Abstract

To date, no study precisely described ear, nose and throat (ENT) disease in adults with primary ciliary dyskinesia (PCD) and its relationship with ciliary function/ultrastructure. A retrospective study of standardized ENT data (exam, audiogram, sinus Computed tomography (CT), and bacteriology) was conducted in 64 adults with confirmed PCD who were followed in two ENT reference centers. Rhinorrhoea and hearing loss were the main symptoms. Symptom scores were higher in older patients. Nasal endoscopy was abnormal in all patients except one, showing nasal polyps in one-third of the patients and stagnant nasal mucus secretions in 87.5% of the patients. Sinus CT opacities were mainly incomplete and showed one-third of the patients with sinus hypoplasia and/or agenesis. Middle meatus mainly grew *Haemophilus influenzae*, *Streptoccocus pneumoniae* and *Pseudomonas aeruginosa*. Otitis media with effusion (OME), which is constant in childhood, was diagnosed in less than one-quarter of the patients. In two-thirds of the patients, audiogram showed hearing loss that was sensorineural in half of the patients. ENT disease severity was not correlated with ciliary function and ultrastructure, but the presence of OME was significantly associated with a forced expiratory volume (FEV1) < 70%. Rhinosinusitis is the most common clinical feature of PCD in adults, while OME is less frequent. The presence of active OME in adults with PCD could be a severity marker of lung function and lead to closer monitoring.

## 1. Introduction

Primary ciliary dyskinesia (PCD) is a rare autosomal recessive inherited disorder affecting approximately 1:10,000 to 1:30,000 individuals [1]. The pathophysiology of PCD is characterized by defects in airway mucociliary clearance, predisposing affected individuals to recurrent respiratory infections and leading to symptoms typically present since birth [2]. Approximately half of sufferers have situs inversus (SI) [3,4] and a majority of male patients are infertile [5]. One subgroup of patients with PCD have an array of chronic rhinosinusitis, bronchiectasis, and SI, called Kartagener syndrome (KS).

Otologic presentation has been well described in children [4,6,7]. Early in childhood, the main clinical feature is severe and binaural otitis media with effusion (OME) leading to hearing impairment and causing delays in speech acquisition. Otologic disease tends to improve with age [4] while the sinonasal disease persists and may increase [3,8]. In children, some authors have reported nonspecific rhinological symptoms such as nasal discharge and episodic facial pain resulting in chronic rhinosinusitis [3]. Hypoplasia and agenesis of paranasal sinus have also been described.

To date, no study has precisely assessed ear, nose and throat (ENT) disease, clinical features, and the relationship between ENT manifestations and ciliary function/ultrastructure in adults with PCD. In this study, we aimed to evaluate the upper airway phenotype by describing ENT disease in adults with PCD, who were followed in two French reference ENT centers.

## 2. Experimental Section

### 2.1. Patients and Methods

We performed a retrospective study in PCD patients followed in two ENT departments (Mondor Hospital and Intercommunal of Creteil) both of which are members of the French National Reference Centre for Rare Respiratory Diseases (RespiRare).

### 2.2. Study Population

Inclusion criteria was a confirmed diagnosis of PCD and age >18 years at the start of the ENT followup in the reference centers. All patients had bronchiectasis on a computed tomography (CT) scan. Patients with other pathological conditions, such as cystic fibrosis (CF) or immunodeficiency, were excluded. We considered that a patient had a confirmed diagnosis of PCD when at least one or more of the following criteria was present: (i) a clinical diagnosis of KS, (ii) evidence of PCD-specific ciliary ultrastructural defects, and (iii) identification of non-ambiguous causal mutations in a known PCD gene. All patients had measurements of the ciliary beat frequency (CBF) and analysis of ciliary ultrastructure as described previously [9,10,11]. Nasal nitric oxide (nNO) was also measured as previously described [12].

The Ethics Review Board of RespiRare approved the retrospective use of the database register on March 20, 2008 (CCTIRS, no.08.015bis).

### 2.3. Data Collection

For each patient, the following variables were recorded at the time of the first ENT visit in the ENT reference center: demographic data (age, sex, geographical origin, active smoker), neonatal respiratory distress, abnormalities of organ lateralization, fertility disorders, and other features potentially related to PCD (retinitis pigmentosa, polycystic kidney, obesity, and psychomotor retardation). All declarative history of consanguinity, and familial PCD, ENT or respiratory disease was noted. Lung characteristics included symptoms, a history of bronchiectasis and/or asthma, and lung surgery. Results of forced expiratory volume after one second (FEV_1_), nNO, CBF, ciliary ultrastructure and genetic molecular analyses were recorded.

### 2.4. Standardized ENT Evaluation

Throughout the study period, an ENT evaluation was performed during a routine visit in our reference center by the same two senior investigators which included a systematic microscopic examination of the ears and a nasal endoscopy. The clinical ENT observed data were reported during the first visit included nose (i.e., congestion, rhinorrhea, anosmia, hyperreactivity, and pain) and ear (i.e., hearing loss, earache, tinnitus, and otorrhoea) symptoms. Six nasal functional criteria were determined: nasal congestion, rhinorrhea, dysosmia, nasal hyperactivity (itching and pruritus), facial pain, and epistaxis. The severity of each symptom was evaluated by the senior investigator according to a 4-point semi-quantitative scale: 0, no symptoms; 1, moderate symptoms; 2, mild symptoms that slightly interfere with daily activities or sleep; and 3, severe symptoms that severely interfere with daily activities or sleep. For each patient a global functional score was calculated by adding the scores of the 6 functional criteria, with a maximum of 18 [13,14]. A standardized nasal endoscopic evaluation was performed by a single clinician during the routine visit including the following findings: the presence of erythema, polyps, mucosal oedema, secretion discharge (purulent or not), and crusting. In the case of purulent nasal secretions (patients were not systematically symptomatic or having an exacerbation), a sample was obtained under endoscopic view by aspiration in the middle meatus with a sterile tube in order to perform a bacteriological analysis. A pure tone audiogram was performed in patients with an abnormal eardrum and/or complaining of hearing loss. All tested patients underwent tympanometry and audiometric tests which including pure tone audiometry (PTA) with both air and bone conduction at frequencies of 250, 500, 1000, 2000, 4000, and 8000 Hertz (Hz). In the case of a hearing loss that was 20 dB or less, hearing was considered normal or subnormal. In the case of a hearing loss that was greater than 20 dB, we classified hearing loss into 3 groups: sensorineural, conductive, or mixed hearing loss. Sensorineural hearing loss was defined as having equal air and bone conduction decrease thresholds greater than 20 dB. Air-bone gaps greater than 10 dB were considered as conductive or mixed hearing loss. ENT treatments before diagnosis and during followup in the center were listed.

### 2.5. Sinonasal CT Evaluation

Data from CT scans of the paranasal sinuses (when available) included: (i) the Lund–Mackay CT score [15] modified according to the prevalence of sinus aplasia [15], and (ii) the degree of pneumatization of each paranasal sinus according to radiological criteria recently used in adults with cystic fibrosis [16]. The degree of pneumatization of each paranasal sinus was evaluated by the same radiologist.

### 2.6. Statistical Analyses

All analyses were performed using the statistical software package (Statview 4, SAS Institute, Grenoble, France).

We determined the ENT characteristics of the patients at the first visit in our center. The quantitative variables were expressed as a mean (SD) or median (range) when appropriate and the qualitative variables were expressed as a number (percentage).

At the first ENT visit in our reference center, nasal disease (mean overall nasal score, sinonasal CT score) and ear disease (presence of OME) were each separately compared to the lung function and PCD diagnostics features (nNO values, CBF, the ultrastructural phenotype)

Using the unpaired t-test, we compared the quantitative data (i.e., mean overall nasal score and CT score) between two groups according to the FEV1 (FEV_1_ < 70%, or FEV_1_ ≥ 70%), or to the nNO values (<77 nL/min and ≥77 nL/min), or to the CBF (normal CBF (>8 Hz) or abnormal CBF (defined as complete immotility or low CBF < 8 Hz) The mean overall score of nasal symptoms at diagnosis was also compared between two groups (<40 vs ≥40 years) using the unpaired t-test.

A comparison of quantitative data according to the ultrastructural phenotype, which was classified in three groups, (i.e., no detectable defect, CC = Central complex abnormalities; DA = dynein arms abnormalities) respectively, was made using an analysis of variance (ANOVA).

The Chi-Squared statistic was used for testing relationships between the presence (or absence) of OME and: FEV_1_ (<70% or ≥70%), age (<40 vs ≥40 years), nNO (<77 nL/min and normal nNO ≥77 nL/min), CBF (normal or abnormal), or the ultrastructural phenotype.

The correlation between age and nasal symptoms was studied using the Spearman’s test.

Statistical significance was accepted for *p* < 0.05.

## 3. Results

### 3.1. Study Population

Sixty-four adult patients (40 males and 24 females) with a confirmed diagnosis of PCD, from 59 independent families, were included. In the case of typical KS with SI and/or neonatal respiratory distress, the mean age (±SD) at diagnosis was 16.5 ± 15 years. For the remaining patients, the mean age at diagnosis was 30 ± 14 years. Patient characteristics are reported in Table 1. The mean age (±SD), at the first ENT visit in the center was 32 ± 11 years. The median value of nNO (range) was 16 (1–634) nL/min and 84.4% of patients had an nNO lower than 77 nL/min (Appendix A). In three patients, n NO was >200 nL/min. Transmission electron microscope (TEM) analysis was performed in 62 patients and ultrastructural defect was identified in 90.3% of them (Appendix A). Molecular analysis was available in 58 patients and causal biallelic mutations in a known PCD gene were identified in 77% of them. Lung characteristics are described in Table 1. Bronchiectasis were present in all patients (100%). A FEV_1_ < 70% was reported in 50% of patients.

### 3.2. Description of the Sinonasal Disease

Sixty-three patients (98.4%) had a history of recurring problems at the paranasal sinuses (Table 2). All patients had at least one sinonasal symptom (Table 2). Rhinorrhoea was the most frequent symptom (*n* = 61 or 95.3% of the patients). At diagnosis, the mean overall score of nasal symptoms was 5.6 ± 2.7 (media *n* = 6, range 1–12) and higher in the ≥40-year subgroup (mean 6.5 ± 3.1) compared to the <40-year subgroup (mean 5 ± 2.3) (*p* = 0.04, student). There was a significant correlation between age and nasal symptom score (*r* = 0.2, *p* = 0.02, Spearman) (Figure 1). Only one patient had a normal nasal endoscopic examination. The most frequent endoscopic abnormalities were oedema and congestion of nasal mucosa (*n* = 9 or 14% of the patients and *n* = 28 or 43.7% of the patients, respectively). Polyps were observed in 22 patients (34.4%). Stagnant nasal mucus secretions were present in 56 patients (87.5%) with purulent secretions in 55.3% of them. In 35 patients (54.7%) we observed sticky secretions that were strung between the septum and the inferior turbinates (Figure 2). Bacteriological culture of the middle meatus was performed in 31 patients and showed one or more bacteria in 83.9% (*n* = 26). The most common bacteria were *Haemophilus influenza* (25.8%), *Pseudomonas aeruginosa* (19.3%), and *Streptococcus pneumonia* (19.3%). A sinus CT scan was available for 41 patients showing agenesis and/or hypoplasia (mainly frontal sinus) in 15 patients (36.6%) (Figure 1). Sinus opacities were present in 36 patients (89%) (Figure 1). A normal sinus CT scan was observed in five patients (12.2%). The mean CT score was 6.2 ± 3.6. No complications such as mucocele were reported. Thirty-seven patients (57.8%) had a history of sinonasal surgery before the first ENT visit to the reference centre. Sinonasal treatments at diagnosis are reported in Table 2. The main nasal treatments initiated after diagnosis included meticulous nasal rinsing with daily isotonic saline (twice daily) for 83% of the patients and local corticosteroids for 62% of the patients. After diagnosis, sinonasal surgical treatment was performed in 19% of the patients. Surgery concerned the ethmoidal and/or frontal sinuses (40%), maxillary sinuses (10%), turbinates (35%), polyps (5%), and septum (5%).

### 3.3. Description of the Ear Disease

Along with the sinonasal disease, a history of OME was reported in 58 patients (90.7%) and a history of ventilation tube insertion during childhood was reported in 24 patients (37.5%) (Table 3). One or more ear symptoms were present in 43 patients (67.2%) with hearing loss being the most frequent (*n* = 34 or 53.1% of the patients). Upon examination, 19 patients (29.6%) had normal bilateral eardrums. At least one abnormal eardrum was observed in 45 patients (70.3%) (i.e., active OME or sequela of chronic otitis such as tympanic retraction, tympanosclerosis, perforation). Active OME was present in only 24% of eardrums. A pure tone audiogram was performed in 45 patients and hearing loss (air conduction ≥10 dB) was observed in 30 patients (66.7%). Among these, 46.7% had a pure conductive hearing loss, 16.7% had a pure sensorineural hearing loss, and 36.6% had a combined hearing loss. No patient had ear surgery (ventilation tube insertion or tympanoplasty) or hearing aids after PCD diagnosis.

### 3.4. Correlation between Upper Airways Disease and Characteristics of PCD

There wasn’t any significant difference of the mean overall score of nasal symptoms according nNO (*p* = 0.6), CBF (*p* = 0.5), ultrastructural phenotypes (*p* = 0.12) or lung function (*p* = 0.94).

Neither was any significant difference for the CT score according to the age (<40 vs ≥40 years) (*p* = 0.85), CBF(>8 Hz or <8 Hz) (*p* = 0.76), ultrastructural phenotype (*p* = 0.5) lung function (FEV_1_ < 70% or ≥70%) (*p* = 0.86), or nNO (*p* = 0.61).

In particularly, nasal disease (mean overall score of nasal symptoms and CT score) was more severe in the TEM group (absence of inner dynein arms with disorganization) than other groups (CC abnormalities and no detectable defect) without reaching significant difference.

No correlation was found between the presence of OME and age (*p* = 0.99), CBF (*p* = 0.36), ultrastructural phenotype (*p* = 0.16).

There was significantly more OME in patients with FEV_1_ < 70% (*p* = 0.01).

## 4. Discussion

To our knowledge, this is the largest study describing ENT disease in patients who started their follow-up in a PCD reference center as adults. Nearly all the patients had ENT symptoms since childhood, mainly rhinorrhoea and hearing loss. The nasal symptom score was higher in patients over 40 years of age. Nasal endoscopy was abnormal in all patients except one and it revealed chronic rhinosinusitis with nasal polyps in a third of the patients. Nearly a third of the patients had normal eardrums, however, OME, which is consistently seen in childhood, was diagnosed in less than a quarter of the adults. Hearing loss (either conductive or sensorineural) was found in two-thirds of the patients who underwent an audiogram. ENT disease severity was not correlated to any characteristics of PCD (i.e., nNO, CBF, and ultrastructural phenotype). However, OME in adults could be a severity marker of lung function.

One limitation of our study is the retrospective collection of data, which may have introduced a bias in evaluating ENT disease in our study population. However, a standardized case report form was systematically completed by each senior investigator. Some complementary investigations such as CT scan, audiogram, and bacteriological culture were not systematically performed.

Sinonasal presentation of the disease in adults was similar to that in children but worsened with age. A history of chronic rhinosinusitis was always described in our PCD population. Only one patient had no previous history of rhinosinusitis and exhibited a normal nasal endoscopy (i.e., a 32-year-old man with hypofertility). In this patient, the diagnosis of PCD was confirmed by identification of a typical ultrastructural defect of cilia (absence of both dynein arms) and of causal mutations in the *DNAAF 1* gene.

Almost all of the patients had an abnormal nasal endoscopy corresponding to chronic rhinosinusitis, even those with few symptoms. These findings contrast with the retrospective study by Noone et al. who reported sinusitis in 47% of adult patients with PCD [17]. However, their diagnosis was based only on previous sinus surgery or sinus radiographs. Another study of 44 adult patients with PCD, based on an auto-questionnaire to determine ENT clinical manifestations (with only a response rate of 46%) reported recurring problems at the paranasal sinuses in 59% of all patients, and 69% of these patients needed corresponding surgical intervention [18]. A systematic nasal endoscopic examination, as performed in our ENT clinic, provides a more objective approach. A typical nasal sign in endoscopy is the presence of sticky secretions strung between the septum and the inferior turbinate. Nasal polyps in PCD have already been reported in the literature as occurring in 25% to 52% of adults [11,19] which is similar to our findings. The prevalence of nasal polyps in PCD is similar to cystic fibrosis (prevalence of 39–48%) [20,21]. Nasal polyps in PCD differ from the classic eosinophilic sinonasal polyposis, since they are associated with oedema, congestive mucosa, and sticky or purulent secretions. Such an atypical subtype of rhinosinusitis is mainly found in rare conditions such as cystic fibrosis and immunodeficiency which represent the classical differential diagnosis of PCD [22,23].

Sinus CT scan analysis showed sinus hypoplasia and/or aplasia in around a third of our patients, which is lower than in the study by Pifferi et al. and El Sayed et al. (73% and 56% of abnormalities, respectively) [24,25]. As previously reported, sinus opacities were found in almost all our patients [26] but adults with PCD can show a normal sinus CT scan as described in our study.

However, sinus CT opacities were mainly incomplete. Consequently, as already described [17], the mean CT score was low, which contrasts with the reported score in non-PCD patients with chronic sinusitis and asthma [27].

Ear disease appears to be less severe in adults with PCD than in children. OME, which is present in nearly all children with PCD [4], was found in approximately one-quarter of our adult patients. This contrasts with a retrospective study by Anderson et al. [28] who found 68% of ears with OME in the age group >17 years (*n* = 33) but the mean (interval) age of the group was not given and could introduce a bias [28]. Ear disease would thus appear to be less frequent than sinonasal disease in adults. In our study, unlike children, very few adults complain of otorrhoea and earache which is in contrast with a study by Anderson et al. [28]. This difference could be explained by a higher percentage of patients with ventilation tubes the study by Anderson et al. than in our study (54% versus 37.5%, respectively). In our adult study, hearing loss measured with audiometry was present in half of the patients, which was in line with a previous questionnaire-based study [18]. Anderson et al. [28], noted that hearing significantly improved with increasing age, but considering their results in detail it showed that only air conduction improved while bone conduction worsened, as found in our study. Indeed, in our study, we found that the subtype of hearing loss is different in adults as compared with children. Adults tend to have sensorineural hearing loss (as in one third of the patients in our study) while children with PCD mainly have conductive hearing loss secondary to OME. The pathogenesis of sensorineural hearing loss in the adults is not clear. Presbyacusis was only observed in one patient who was 60 years old. Ototoxic adverse effects of routine PCD therapies such as aminoglycoside or macrolides could be involved, or the passage of bacterial toxins into the inner ear [29]. Lundman et al. demonstrated in chinchillas that *Pseudomonas aeruginosa* exotoxin A can cross the round window, causing irreversible hair cell loss [30]. Moreover, despite chronic otitis in childhood, no complications such as cholesteatoma were reported in our PCD cohort as in the literature [25] probably because of a different physiopathology. Interestingly, a significant correlation was found between the presence of OME and lung function in our study. This result needs to be confirmed but it could suggest that OME in an adult with PCD could be a severity marker of lung function and lead to closer follow-up and more aggressive treatment.

As compared with lower airway disease in PCD, the natural course of upper airway disease is less known in adults. We observed a significant worsening of sinusitis with age. For most patients, nasal surgery was performed early in the natural history of their ENT disease, and mainly before PCD diagnosis. In a recent adult study, 52% of patients had sinus surgery at least once [19]. When the PCD diagnosis was established, the type of sinonasal surgery focused on the inferior turbinates and ethmoid to improve nasal obstruction and rhinorrhoea [31]. It is important to note that sinonasal surgery could complicate nasal sampling for ciliary analysis and thus delay PCD diagnosis. After PCD diagnosis, very few patients underwent ENT surgery for chronic rhinosinusitis and none for ear disease.

This study highlights that ENT data could be used to propose a new specific adult clinical score to select patients for PCD diagnostic testing [32]. Indeed, the clinical score used in pediatric populations is probably less appropriate in adults. Moreover, retrospective clinical information data collected in adults are less relevant than data collected in children, especially concerning neonatal respiratory distress [33,34].

## 5. Conclusions

This study has resulted in a change of daily practice in our centers. Firstly, in adults suffering from ENT symptoms since childhood, signs suggestive of PCD are: the presence of thick nasal secretion with or without polyps, OME, and hypoplasia or agenesis on the CT scan (even if sinus ventilation is normal or the CT score of sinus opacities is low). These signs are suggestive of a ciliary disorder and worth investigating. Secondly, considering the high proportion of sensorineural hearing loss, we systematically perform temporal bone imaging at diagnosis and audiometric follow-up, which can be intensified if the patient is prescribed potentially ototoxic antibiotics. Thirdly, our follow-up focuses more specifically on sinonasal symptoms control as a longitudinal reflection of sinusitis symptomatology, which is the primary cause of decreased quality of life and the primary outcome measure used for clinical decision-making. In our experience, nasal endoscopy is a secondary outcome measure (used for nasal obstruction evaluation and/or bacterial sampling of the middle meatus) and CT scan is never used as a follow-up exam.

This study opens a wide field of ENT-guided studies in order to: (i) improve PCD diagnosis in adults by establishing a screening score; (ii) to improve ENT management and follow-up (sinonasal surgery indications, monitoring of adult patients with OME, internal ear explorations); and (iii) to understand the relationship between ENT and pulmonary diseases.

## Figures and Tables

**Figure 1 jcm-08-00619-f001:**
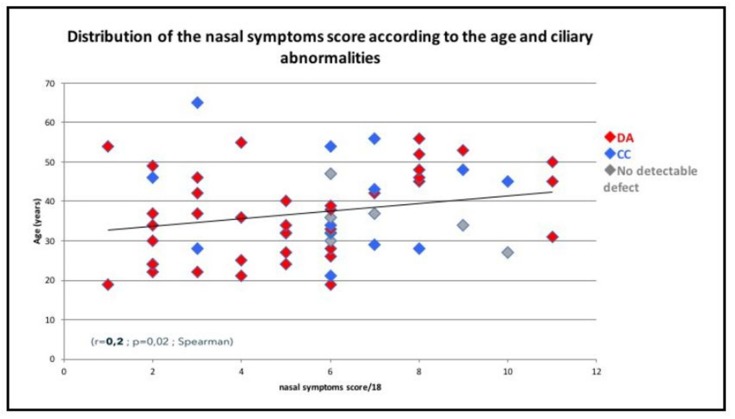
Distribution of nasal symptoms score according to the age and ciliary abnormalities. Abbreviations: CC = central complex abnormalities; DA = dynein arms abnormalities.

**Figure 2 jcm-08-00619-f002:**
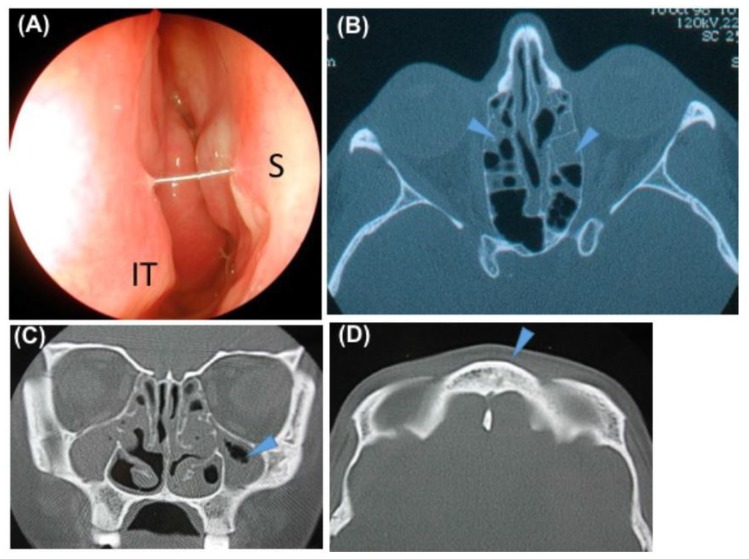
Typical nasal endoscopy and sinus computed tomography (CT) scan in an adult with primary ciliary dyskinesia PCD. (**A**) Typical nasal endoscopy (right side): secretions were sticky and thick, strung between the septum (S) and the inferior turbinate (IT). In a typical form, we observe oedema and congestive mucosa with or without polyps. (**B**) Sinus CT scan axial view: partial opacities of ethmoid cells. (**C**) Sinus CT scan coronal view: maxillary sinus hypoplasia. (**D**) Sinus CT scan axial view: frontal sinus agenesis.

**Table 1 jcm-08-00619-t001:** Primary ciliary dyskinesia (PCD) patient characteristics (*n* = 64, except when specified).

Characteristics				
			Mean	±SD
Age at the first ENT visit in our center (years)			32	11
			Patient (*n*)	%
Gender				
Female			24	37.5
Male			40	62.5
PCD related diseases				
Neonatal respiratory distress			18	28.1
Situs inversus			22	34.3
Heterotaxy			4	6.2
Fertility disorders			36	56.2
Retinitis pigmentosa			3	4.7
Polycystic kidney disease			2	3.1
Family history (*n* = 59 independent families)			Families (*n*)	
Consanguinity			18	30.5
PCD *			11	18.6
Situs inversus			4	6.8
Rhinosinusitis			15	25.4
Otitis			9	15.2
Asthma			9	15.2
Bronchiectasis			9	15.2
Lung diseases				
Asthma			14	22
Bronchiectasis			64	100
Lung surgery				
Lobectomy			15	23.4
Transplantation			1	1.6
Lung function test *n* = 45				
FEV1/FVC% mean ± SD (range)			73.6 ± 13.8 (47–100)	
FEV1% mean ± SD (range)			72.7 ± 24.7 (18–128)	

Abbreviations FEV1 = forced expiratory volume; FVC = forced vital capacity; SD = standard deviation. * Family history of PCD: All the other family members who had PCD were siblings except one case (the other family member who also had PCD was the mother of the patient).

**Table 2 jcm-08-00619-t002:** Sinonasal disease characteristics at diagnosis in our reference center (*n* = 64).

	Patient (n)	%	
History of Sinonasal Surgery before the First ENT Visit in our Reference Center			
Unspecified surgery	22	34.4	
Middle meatotomy	9	14.1	
Ethmoidectomy	8	12.5	
Turbinate reduction	8	12.5	
Polypectomy	1	1.6	
Sinonasal symptoms			Score of severity *: mean ± SD (range)
Rhinorrhoea	61	95.3	1.9 ± 0.8 (0–3)
Congestion	42	65.6	1.3 ± 1.1 (0–3)
Facial pain	35	54.7	1 ± 0.9 (0–3)
Dysosmia	35	54.7	1 ± 1.19 (0–3)
Nasal hypereactivity	9	14.1	0.2 ± 0.6 (0–3)
Epistaxis	4	6.2	0 ± 0.2 (0–1)
Nasal endoscopy			
Inferior turbinate			
Normal	33	51.6	
Hypertrophy	22	34.4	
Atrophy/scar	10	15.6	
Nasal mucosa			
Normal	13	20.3	
Congestion	28	43.7	
Œdema	9	14	
Polyps	22	34.4	
Nasal secretions			
Normal	8	12.5	
Sticky	35	54.7	
Purulent	21	32.8	

Legends: * The severity of each symptom was evaluated according to a 4-point semi-quantitative scale: 0, no symptoms; 1, moderate symptoms; 2, mild symptoms that slightly interfere with daily activities or sleep; and 3, severe symptoms that severely interfere with daily activities or sleep.

**Table 3 jcm-08-00619-t003:** Ear and hearing disease at diagnosis in our reference center (*n* = 64 except when specified).

	*n*	%
Symptoms (*n* = 64 patients)		
Hearing loss	34	53.1
Ear pain	9	14.1
Tinnitus	6	9.4
Otorrhea	5	7.8
Otoscopy (*n* = 128 eardrums)		
Normal	53	41.4
OME	31	24.2
Tympanic retraction	13	10.1
Perforation	8	6.2
Tympanosclerosis	18	14
Ventilation tubes	5	3.9
Audiogram (*n* = 90 ears)		
Normal	33	36.7
Conductive hearing loss	28	31.1
Combined hearing loss	16	17.8
Sensorineural hearing loss	13	14.4

Abbreviations: OME = otitis media with effusion.

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
