# Peer review of "Critical Evaluation of Sinonasal Disease in 64 Adults with Primary Ciliary Dyskinesia"

_jcm, 2019, doi:10.3390/jcm8050619_

Reviewer 1 Report

In this paper, the authors examined ENT data of 64 adult patients with primary ciliary dyskinesia (PCD), which is the largest in the literature. Methods used are appropriate and the results obtained are interesting and will be useful. However, the following points should be reconsidered.

1.    According to the Table 1, 11 families had a family history of PCD. Please clarify what other family members (siblings, parents, etc) had PCD in those 11 families. Since most of the PCD traits are transmitted in an autosomal recessive fashion, I assume that most of them had a family history of PCD in their siblings.

2.    In Figure 1 (C), hypoplastic maxillary sinus is shown. What are the criteria of maxillary sinus hypoplasia? Although hypoplasia of frontal sinuses and sphenoid sinuses are well recognized as a feature of PCD, it is not well known that hypoplasia of maxillary sinus is also a characteristic finding of PCD.

3.    The description of the ear diseases, in the result section, is ambiguous.  I think that authors had better write the mean air conduction thresholds levels measured by pure-tone audiometry, in order to show the severity of hearing loss. The author can cite a report by Takeuchi K, et al (Analysis of otologic features of patients with primary ciliary dyskinesia. Otol Neurotol. 2017 Dec;38(10):e451-e456. ) and compare the hearing levels.

Author Response

Response to Reviewer 1 Comments

 I thank you for taking the time to consider our manuscript and for your pertinent comments 

Yes

Can be   improved

Must be improved

Not   applicable

Does the introduction provide sufficient background   and include all relevant references?

(x)

( )

( )

( )

Is the research design appropriate?

(x)

( )

( )

( )

Are the methods adequately described?

(x)

( )

( )

( )

Are the results clearly presented?

( )

( )

(x)

( )

Are the conclusions supported by the results?

(x)

( )

( )

( )

In this paper, the authors examined ENT data of 64 adult patients with primary ciliary dyskinesia (PCD), which is the largest in the literature. Methods used are appropriate and the results obtained are interesting and will be useful. However, the following points should be reconsidered.

 POINT 1 According to the Table 1, 11 families had a family history of PCD. Please clarify what other family members (siblings, parents, etc) had PCD in those 11 families. Since most of the PCD traits are transmitted in an autosomal recessive fashion, I assume that most of them had a family history of PCD in their siblings.

 Response 1 We agree with your suggestion to precise other family members had PCD in 11 families with family history of PCD. All other family members had PCD were siblings except one case (the other family member had also PCD was the mother of the patient).We added this precision in the table 1 with *:

 * Family history of PCD: All other family members had PCD were siblings except one case (the other family member had also PCD was the mother of the patient).”

POINT 2: In Figure 1 (C), hypoplastic maxillary sinus is shown. What are the criteria of maxillary sinus hypoplasia? Although hypoplasia of frontal sinuses and sphenoid sinuses are well recognized as a feature of PCD, it is not well known that hypoplasia of maxillary sinus is also a characteristic finding of PCD.

We agree with your comment and we were also surprised to observe maxillary sinus hypoplasia in PCD patients. Several attempts to classify the MSH are reported in the literature. The most currently accepted classification has been introduced by Bolger et al. (Bolger WE, Woodruff WW, Morehead J. Parsons D (1990) Maxillary sinus hypoplasia: Classification and description of associated uncinate process hypoplasia. Otolaryngol Head Neck Surg 103: 759-765.) According to this classification maxillary sinus hypoplasia shows three distinct hypoplasia patterns. Type I characteristics are mild

hypoplasia of the maxillary sinus, normal uncinate process and well-developed infundibular passage. Significant hypoplasia of the maxillary sinus, hypoplastic or absent uncinate process and absent or pathologic infundibular passage are seen in Type II. Type III is characterized by the absence of uncinate process and cleft-like maxillary sinus hypoplasia. According to this classification, our exemple seen in figure 1C correspond to a type II maxillary sinus hypoplasia.

POINT 3 The description of the ear diseases, in the result section, is ambiguous.  I think that authors had better write the mean air conduction thresholds levels measured by pure-tone audiometry, in order to show the severity of hearing loss. The author can cite a report by Takeuchi K, et al (Analysis of otologic features of patients with primary ciliary dyskinesia. Otol Neurotol. 2017 Dec;38(10):e451-e456. ) and compare the hearing levels.

Response 3: We agree with your pertinent comment. The description of the ear diseases in the result section is ambiguous because we didn’t precisely define in the method section the design of  collected retrospective data. A pure tone audiogram was performed in patients with abnormal eardrum and/or complaining of hearing loss. In case of hearing  loss of 20 dB or less, hearing was considered normal or subnormal In case of hearing loss more than 20 dB we classified hearing loss into 3 groups Sensorineural, conductive or mixed hearing loss. Sensorineural hearing loss was defined as equal air and bone conduction decrease thresholds greater than 20 dB. Air-bone gaps greater than 10 dB were considered as conductive or mixed hearing loss.  But the degrees of hearing loss and hearing levels at each frequency were not reported in our standardized collection of data because the design of our studies was more focused on chronic rhinosinusitis than on ear disease .

 We added this information in the method section line 120-127:

 “A pure tone audiogram was performed in patients with abnormal eardrum and/or complaining of hearing loss. All tested patients underwent tympanometry and audiometric tests including pure tone audiometry (PTA) with both air and bone conduction at frequencies of 250, 500, 1000, 2000, 4000, and 8000 Hertz (Hz). In case of hearing  loss of 20 dB or less, hearing was considered normal or subnormal In case of hearing loss greater than 20 dB we classified hearing loss into 3 groups Sensorineural, conductive or mixed hearing loss. Sensorineural hearing loss was defined as equal air and bone conduction decrease thresholds greater than 20 dB. Air-bone gaps greater than 10 dB were considered as conductive or mixed hearing loss.”

 Reviewer 2 Report

To the Authors: The authors describe ENT features of a relatively large adult patients with PCD, a group that are both understudied and under diagnosed. There is a wealth of data that could be shared.  Some further details related to the features of data collection, methodologic features, data analysis and the patient status should be included as detailed below:

 1.     ENT scoring system. The authors should provide additional details. For example, has this been previously used, is it validated, and what is the variation between investigators scoring the same patient (intra observer variation).  It would be useful to include the scoring system in a table. I cannot determine the 0-18 points by the test.

 2.     Data comparisons and statistical methods. Line 114-124.  “At the first ENT visit, mean overall nasal score/diagnosis of OME/CT-score were compared to the lung function..” The use of the oblique stroke “slash” is confusing, the use of commas may be clearer and how these were compared should be stated. If multiple variables are compared, then perhaps additional statistical testing should be provided by ANOVA.

 3.     Patient population. Based on the methods (lines 88-94) the patients appear to be fairly well characterized but the features may be heterogeneous and that variation is relevant to the variation in ENT diseases. The variation in PCD features and related ENT symptoms is of great interest to those in the field.  An additional table should be provided characterizing each of patient’s age, sex and specific  diagnostic features should be provided. This could be a separate table, an appendix or supplemental data. Included should be the diagnostic features of PCD: respiratory distress at birth, situs status (solitus, inversus or ambiguous, with details of heterotaxy), TEM, CBF, nNO, age at dx of PCD, and who has PCKD, etc.

 4.     Table 1. The term “Kartagener syndrome” is confusing if 22 patients had KS but only 4 had situs inversus. Overall, this term has fallen out of favor to be replaced globally by PCD with specific details provided. Also, Line 57. “The usual description combines…” would be more correctly stated as “One subgroup of patients with PCD have an array of chronic rhinosinusitis, bronchiectasis and SI, called Kartagener syndrome (KS).

 5.     Figure 2. The disorganization of microtubule pairs associated with CCDC39 and CCDC40 mutations are associated with worse pulmonary function. The authors should state in the text if the CC abnormalities include these mutations or TEM showing microtubular pair disorganization. If not, the authors should indicate in the text that this group of PCD patients was included or not, and the severity of their disease.

 6.     Endoscopic, sinus CT, and otologic examination data—these data need detail as to the reason for the examination and reproducibility. State if these were obtained during an exacerbation or a routine visit. State in lines 148-152 (and methods line 96-105). Likewise, in Results, describing  subjective data regarding appearance of nasal sinus mucosa. The authors should state if examination was prompted by a disease exacerbation or was a routine examination. The authors should also state if the findings were a single or repeat examination and if observed by a single clinician or confirmed by a second. Similar questions are related to CT evaluation (lines 106-110 and 156-158) and the ear exam (lines 180-183), and audiograms. Likewise, state if the bacteriologic data obtained when patients were symptomatic or having an exacerbation.

 7.     The discussion should be re-written to be more concise. A number of important points are lost in the overall length of the text and interposed with additional, and often unnecessary, information.  There are many instances where information from the results section is simply restated. 

 Minor

 8.     Study type: Study design – In the abstract it is stated this is a retrospective study (line 36), however in the ‘Experimental Section’ line 71 states this is a “retrospective observational study.”  By definition an observational study must be prospective. For this study the word “observational” should be deleted.

 9.     If used, a copy of the standardized collection form should included in the supplement.

 10.  Text language and typographical errors:

Line 62. “unspecific”, should be  “non-specific”

Line 93. FEV1 = forced expiratory volume’ to state ‘forced expiratory volume after one second”

Line 272. “aminoside” should be aminoglycoside.

Line 277 insert “of” one case of cholesteatoma

Author Response

Response to Reviewer 2 Comments

 I thank you for taking the time to consider our manuscript and for your pertinent comments

 To the Authors: The authors describe ENT features of a relatively large adult patients with PCD, a group that are both understudied and under diagnosed. There is a wealth of data that could be shared.  Some further details related to the features of data collection, methodologic features, data analysis and the patient status should be included as detailed below:

 POINT 1: ENT scoring system. The authors should provide additional details. For example, has this been previously used, is it validated, and what is the variation between investigators scoring the same patient (intra observer variation).  It would be useful to include the scoring system in a table. I cannot determine the 0-18 points by the test.

Response 1: We agree with your suggestion to precise the ENT scoring system used in our methods.

We added the following explanation in the methods sections:

“Six  functional criteria were determined: Nasal congestion, Rhinorrhea, dyssomnia, nasal hyperactivity (itching and pruritus), facial pain and epistaxis. The severity of each symptom was evaluated according to a 4-point semi quantitative scale: 0, no symptoms; 1, moderate symptoms 2 mild symptoms that slightly interfere with daily activities or sleep; and 3, severe symptoms that severely interfere with daily activities or sleep. For each patient a global functional score was calculated by summing the scores of the 6 functional criteria, with a maximum of 18.”

 Intraobserver variation  has not been tested for this paper but  we added  in the methods some previous references when this ENT scoring was used in others studies, such as:

Laryngoscope. 2001 May;111(5):894-9.  Radiofrequency is a safe and effective treatment of turbinate hypertrophy. PMID:11359172

Coste A1Yona LBlumen MLouis BZerah FRugina MPeynègre RHarf AEscudier E.

We added a more detailed explanation of ENT scoring in the table 2.

Patient (n)

%

History of sinonasal   surgery before the first ENT visit in our reference center

     Unspecified   surgery

22

34.4

     Middle meatotomy

9

14.1

     Ethmoidectomy

8

12.5

     Turbinate reduction

8

12.5

     Polypectomy

1

1.6

Sinonasal   symptoms

Score of severity*: mean±SD (range)

     Rhinorrhoea

61

95.3

1.9±0.8 (0-3)

     Congestion

42

65.6

1.3±1.1 (0-3)

     Facial pain

35

54.7

1±0.9 (0-3)

     Dyssomnia

35

54.7

1±1.19 (0-3)

     Nasal hypereactivity

9

14.1

0.2±0.6 (0-3)

     Epistaxis

4

6.2

0±0.2 (0-1)

Nasal   endoscopy

      Inferior   turbinate

                    Normal

33

51.6

                    Hypertrophy

22

34.4

                    Atrophy/scar

10

15.6

     Nasal mucosa

                    Normal

13

20.3

                    Congestion

28

43.7

                    Œdema

9

14

      Polyps

22

34.4

     Nasal   secretions

                     Normal

8

12.5

                      Sticky

35

54.7

                      Purulent

21

32.8

* The severity of each symptom was evaluated according to a 4-point semi quantitative scale: 0, no symptoms; 1, moderate symptoms 2 mild symptoms that slightly interfere with daily activities or sleep; and 3, severe symptoms that severely interfere with daily activities or sleep.

POINT 2: Data comparisons and statistical methods. Line 114-124.  “At the first ENT visit, mean overall nasal score/diagnosis of OME/CT-score were compared to the lung function..” The use of the oblique stroke “slash” is confusing, the use of commas may be clearer and how these were compared should be stated. If multiple variables are compared, then perhaps additional statistical testing should be provided by ANOVA.

Response 2: We agree with your comment. We modified the sentence using commas:

At the first ENT visit in our reference centre, nasal disease (mean overall nasal score, sinonasal CT score)  and ear disease ( presence of OME)  were each separately compared to the lung function and PCD diagnostics features (nNO values, CBF, the ultrastructural phenotype)”

We also agree with the second part of your comment : The presentation of statistical methods is confusing, thus difficult to understand which statistical test was performed for each comparison. We corrected the statistical analysis paragraph of the methods and added more details in result section including the p values.

We also provided additional statistical ANOVA in order to evaluate correlation of ENT disease with the ultrastructural phenotype (classified into 3 groups).

 We propose you this new version of statistical methods and results:

“Method section

Statistical Analyses

All analyses were performed using a statistical software package (Statview 4; SAS Institute; Grenoble, France). 

We determined the ENT characteristics of the patients at the first visit in our centre: quantitative variables were expressed as mean (SD) or median (range) when appropriate and qualitative variables as number (percentage).

At the first ENT visit in our reference centre,  nasal disease (mean overall nasal score, sinonasal CT score)  and ear disease ( presence of OME)  were each separately compared to the lung function and PCD diagnostics features (nNO values, CBF, the ultrastructural phenotype)

Using the unpaired t-test, we compared the quantitative data (I.e. mean overall nasal score and CT-score) between two groups according to the FEV1 (FEV1 <70%, or FEV1³ 70%), or to the nNO values (<77 nl/min and ≥77 nl/min), or to the CBF (normal CBF (>8Hz) or abnormal CBF (defined as Complete immotility or low CBF<8hz< span="">) The mean overall score of nasal symptoms at diagnosis was also compared between two groups (< 40 vs ≥ 40 years) using the unpaired t-test.

Comparison of quantitative data according to the ultrastructural phenotype classified in three groups, i.e (No detectable defect, CC=Central complex abnormalities; DA=dynein arms abnormalities) respectively, was made using an analysis of variance (ANOVA).

The Chi-Squared statistic was used for testing relationships between the presence (or absence) of OME and: FEV1 (<70%< span=""> or ³ 70%), age (< 40 vs ≥ 40 years), nNO (<77 nl/min and normal n(NO) ≥ 77 nl/min), CBF (normal or abnormal) or the ultrastructural phenotype.

Correlation between age and nasal symptoms was studied using the Spearman’s test.

Statistical significance was accepted for p<0.05.< span="">

 Result section

Correlation between upper airways disease and characteristics of PCD

There wasn’t any significant difference of the mean overall score of nasal symptoms according  n(NO) (p=0.6), CBF (p=0.5) , ultrastructural phenotypes (p=0.12 ) or lung function (p=0.94).

Neither was any significant difference for the CT score according to the age (< 40 vs ≥ 40 years) (p=0,85), CBF(>8Hz or<8hz)< span="">(p=0.76) , ultrastructural phenotype (p=0.5) lung function (FEV1 <70% or ³ 70%) (p=0.86), or n(NO) (p=0.61)

 In particularly, nasal disease (mean overall score of nasal symptoms and CT score) was more severe in the TEM group of “Absence of inner dynein arms with disorganization” than other groups (CC abnormalities and no detectable defect) without reaching significant difference.

No correlation was found between the presence of OME and age (p=0.99), CBF (p=0.36), ultrastructural phenotype  (p=0.16)

There was significantly more OME in patients with FEV1 <70% (p=0.01).

POINT 3: Patient population. Based on the methods (lines 88-94) the patients appear to be fairly well characterized but the features may be heterogeneous and that variation is relevant to the variation in ENT diseases. The variation in PCD features and related ENT symptoms is of great interest to those in the field.  An additional table should be provided characterizing each of patient’s age, sex and specific  diagnostic features should be provided. This could be a separate table, an appendix or supplemental data. Included should be the diagnostic features of PCD: respiratory distress at birth, situs status (solitus, inversus or ambiguous, with details of heterotaxy), TEM, CBF, nNO, age at dx of PCD, and who has PCKD, etc.

Response 3: We agree with your comment to precise diagnostic features of PCD in our population.  Some clinical features of PCD (respiratory distress at birth, situs inversus, PCKD, etc…) are already in the table 1 but we added in the table 1 epidemiologic characteristics of our patients (sex, age) but results of PCD diagnostic tests are lacking (NO, CBF, TEM). We then added  these supplementary  data in a  supplementary  table :

Table S1:    PCD diagnostic tests 

mean

(± SD)

Age at diagnosis   of patients with SI and/or respiratory distress (years)

16.5

15

Age at diagnosis   of other  patients (years)

30

14

Patient   (n)

%

 (NO) (nl/min) (n=45)*

       n(NO)<77 nl/min

38

84.4

       n(NO) ≥77 nl/min

7

15.6

Ciliary   beat  frequency (CBF) (n=59)**

       Complete immotility

40

67.9

       Low CBF<8hz< span="">

10

17

       CBF>8Hz

8

13.3

       Not evaluable

1

1.8

Ciliary   ultrastructure (n=62)

       Absence of outer dynein arms

17

27.4

       Absence of both dynein arms

14

22.6

       Central complex abnormalities

14

22.6

       Absence of inner dynein arms with disorganisation

11

17.7

       No detectable defect***

6

9.7

  Causal biallelic mutation

50

77

* Cut- off determined by Jackson CL, Behan   L, Collins SA, et al. Accuracy of diagnostic testing in primary ciliary   dyskinesia. The European Respiratory Journal. 2016;47(3):837-848.

** During the20 year’study, technical   modifications were introduced for the evaluation of ciliary beat frequency   measurements (Papon J-F, Bassinet L, Cariou-Patron G, Zerah-Lancner F, Vojtek   A-M, Blanchon S, et al. Quantitative analysis of ciliary beating in primary   ciliary dyskinesia: a pilot study. Orphanet J Rare Dis. 2012;7:78). We   considered 3 groups: dyskinetic cilia with low CBF (<8hz), dyskinetic="" cilia="" with="" normal="" cbf="">8Hz), or complete immotility when almost all cilia   are immotile

*** All patients presented Kartagener   syndrome

Abbreviations SI= Situs Inversus, SD=Standard Deviation n(NO)   =Nasal nitric oxide, CBF = Ciliary Beat Frequency

POINT 4: Table 1. The term “Kartagener syndrome” is confusing if 22 patients had KS but only 4 had situs inversus. Overall, this term has fallen out of favor to be replaced globally by PCD with specific details provided. Also, Line 57. “The usual description combines…” would be more correctly stated as “One subgroup of patients with PCD have an array of chronic rhinosinusitis, bronchiectasis and SI, called Kartagener syndrome (KS).

Response 4:  Table 1. We are agree with your comment. KS is confusing in the table one. Therefore, we changed the term “KS” and replaced it by “Situs Inversus”  in Table 1.

“4” is the number of SI in the family history.

Characteristics

mean

+/-   SD

Age at the first   ENT visit in our centre (years)

32

         11

Patient   (n)

%

Gender

     female

24

37.5

     Male                                          

40

62.5

PCD related diseases

     Neonatal   respiratory distress

18

28.1

     Situs Inversus

22

34.3

       Heterotaxy

4

6.2

     Fertility   disorders

36

56.2

     Retinitis   pigmentosa

3

4.7

       Polycystic kydney disease

2

3.1

Family history (n=59   independent families)

Families (n)

      Consanguinity

18

30.5

     PCD*

11

18.6

     Situs   inversus

4

6.8

       Rhinosinusitis

15

25.4

     Otitis

9

15.2

     Asthma

9

15.2

       Bronchiectasis

9

15.2

Lung diseases

     Asthma

14

22

       Bronchiectasis

64

100

Lung surgery

     Lobectomy

15

23.4

       Transplantation

1

1.6

Lung function test  n=45

FEV1/FVC % Mean± SD (range)

73.6± 13.8 (47-100)

FEV1   % Mean± SD (range)

72.7± 24.7 (18-128)

Abbreviations    FEV1= forced expiratory volume; FVC = forced vital capacity.

 We agree with your suggestion and we modified the sentence line 57 as suggested:

“One subgroup of patients with PCD have an array of chronic rhinosinusitis, bronchiectasis and SI, called Kartagener syndrome (KS).”

 POINT 5 Figure 2. The disorganization of microtubule pairs associated with CCDC39 and CCDC40 mutations are associated with worse pulmonary function. The authors should state in the text if the CC abnormalities include these mutations or TEM showing microtubular pair disorganization. If not, the authors should indicate in the text that this group of PCD patients was included or not, and the severity of their disease.

Response 5: Figure 2 CCDC39 and CCDC40 corresponded in all cases to  the TEM group “Absence of inner dynein arms with disorganisation”. Therefore they are not included in the CC abnormalities group.

However as you suggested,  we looked whether nasal disease  (nasal symptoms and CT score) was more severe in the group of “Absence of inner dynein arms with disorganization” than other groups and we found no statistical difference. We added this information in the result section line 214:

“Nasal disease (mean overall score of nasal symptoms and CT score) was more severe in the TEM group of “Absence of inner dynein arms with disorganization” than other groups (CC abnormalities and No detectable defect) without reaching significant difference.”

POINT 6: Endoscopic, sinus CT, and otologic examination data—these data need detail as to the reason for the examination and reproducibility. State if these were obtained during an exacerbation or a routine visit. State in lines 148-152 (and methods line 96-105). Likewise, in Results, describing  subjective data regarding appearance of nasal sinus mucosa. The authors should state if examination was prompted by a disease exacerbation or was a routine examination. The authors should also state if the findings were a single or repeat examination and if observed by a single clinician or confirmed by a second. Similar questions are related to CT evaluation (lines 106-110 and 156-158) and the ear exam (lines 180-183), and audiograms. Likewise, state if the bacteriologic data obtained when patients were symptomatic or having an exacerbation.

Response 6 : We agree with the lack of precision in ENT evaluation (clinical and CT evaluation) in the method section. We added some data in the method section

Reason for examination: all clinical exams were performed during routine visit in our national reference center and therefore the were not performed simultaneousely by two clinicians. The single clinician that performed the clinical exam was a senior investigator. ENT data ( clinical (nose and ear), CT scan, audiogram, and bacteriological data) reported in our study are data from the first ENT visit in our centre. All examinations were performed  by the same two senior investigator  using a standardized nasal evaluation including following findings: the presence of mucosal erythema, polyps, mucosal Oedema, secretion discharge (purulent or not) and crusting.

Concerning the CT evaluation, the degree of pneumatisation of each paranasal sinus was evaluated by the same radiologist. Sinus aplasia or hypoplasia were defined according to radiological criteria recently used in adults with cystic fibrosis (CF)

Concerning the bacteriologic exams, they were performed when purulent secretions were observed by ENT clinician but patients were not systematically symptomatic or having an exacerbation.

Concerning audiometric evaluation, all tested patients underwent tympanometry and audiometric tests including pure tone audiometry (PTA) with both air and bone conduction at frequencies of 250, 500, 1000, 2000, 4000, and 8000 Hertz (Hz). Types of hearing loss (conductive or sensorineural), degrees of hearing loss and hearing levels at each frequency were also recorded. Sensorineural hearing loss was defined as equal air and bone conduction decrease thresholds greater than 20 dB. Air-bone gaps greater than 10 dB were considered as conductive or mixed hearing loss.

We added these details about our standardized ENT evaluation in the method section (line 102-132):

“Standardized ENT evaluation

ENT evaluation was performed during routine visit in our reference centre by the same two senior investigator during all the study period and included systematic microscopic examination of the ears and a nasal endoscopy. The clinical ENT data reported were data observed during the first visit included nose (i.e. congestion, rhinorrhea, anosmia, hyperreactivity, and pain) and ear (i.e. hearing loss, earache, tinnitus and otorrhoea) symptoms. Six nasal functional criteria were determined: nasal congestion, rhinorrhea, dysosmia, ,nasal hyperactivity (itching and pruritus), facial pain and epistaxis. The severity of each symptom was evaluated by senior investigator according to a 4-point semi quantitative scale: 0, no symptoms; 1, moderate symptoms 2 mild symptoms that slightly interfere with daily activities or sleep; and 3, severe symptoms that severely interfere with daily activities or sleep. For each patient a global functional score was calculated by summing the scores of each symptom, with a maximum of 18 [13]. A standardized nasal endoscopic evaluation was performed by a single clinician during the routine visit including following findings: the presence of erythema, polyps, mucosal oedema, secretion discharge (purulent or not) and crusting. In the case of purulent nasal secretions (patients were not systematically symptomatic or having an exacerbation), a sample was obtained under endoscopic view by aspiration in the middle meatus with a sterile tube in order to perform a bacteriological analysis. A pure tone audiogram was performed in patients with abnormal eardrum and/or complaining of hearing loss. All tested patients underwent tympanometry and audiometric tests including pure tone audiometry (PTA) with both air and bone conduction at frequencies of 250, 500, 1000, 2000, 4000, and 8000 Hertz (Hz). In case of hearing  loss of 20 dB or less, hearing was considered normal or subnormal In case of hearing loss greater than 20 dB we classified hearing loss into 3 groups Sensorineural, conductive or mixed hearing loss. Sensorineural hearing loss was defined as equal air and bone conduction decrease thresholds greater than 20 dB. Air-bone gaps greater than 10 dB were considered as conductive or mixed hearing loss. ENT treatments before diagnosis and during follow-up in the centre were listed.

Sinonasal CT evaluation

Data from CT scans of the paranasal sinuses (when available) included: i/ the Lund–Mackay CT score [14] modified according to the prevalence of sinus aplasia [14] and, ii/ the degree of pneumatisation of each paranasal sinus according to radiological criteria recently used in adults with cystic fibrosis [15]. The degree of pneumatisation of each paranasal sinus was evaluated by the same radiologist.”

POINT 7.     The discussion should be re-written to be more concise. A number of important points are lost in the overall length of the text and interposed with additional, and often unnecessary, information.  There are many instances where information from the results section is simply restated. 

 Response 7: We totally agree with your pertinent comment. We proposed a more concise version of the discussion:

“To our knowledge, this is the largest series describing ENT disease in patients started their follow-up in a PCD Reference Centre as adults. Nearly all the patients had had ENT symptoms since childhood, mainly rhinorrhoea and hearing loss. The nasal symptom score was higher in patients over 40 years. Nasal endoscopy was abnormal in all patients except one and revealed chronic rhinosinusitis with nasal polyps in a third. Nearly a third of the patients had normal eardrums but OME, which is consistently seen in childhood, was diagnosed in less than a quarter of the adults. Hearing loss (either conductive or sensorineural) was found in 2/3 of the patients who underwent an audiogram. ENT disease severity was not correlated to any characteristics of PCD (i.e. n(NO), CBF and ultrastructural phenotype). However, OME in adults could be a severity marker of lung function.

One limitation of our study is the retrospective collection of data, which may have introduce bias in evaluating ENT disease in our study population. However, a standardized case report form was systematically completed by each senior investigator. Some complementary investigations such as CT scan, audiogram, bacteriological culture were not systematically performed.

Sinonasal presentation of the disease in adults was similar to that in children but worsened with age. A history of chronic rhinosinusitis was always described in our PCD population. Only one patient had no previous history of rhinosinusitis and exhibited a normal nasal endoscopy: a 32-year-old man with hypofertility. In this patient, the diagnosis of PCD was confirmed by identification of typical ultrastructural defect of cilia (absence of both dynein arms) and of causal mutations in the DNAAF 1 gene.

Almost all of the patients had an abnormal nasal endoscopy corresponding to chronic rhinosinusitis, even those with few symptoms. These findings contrast with the retrospective cohort of Noone et al. who reported sinusitis in 47% of adult patients with PCD [16]. However, their diagnosis was based only on previous sinus surgery or sinus radiographs. Another study of 44 adult patients with PCD, based on an auto-questionnaire to determine ENT clinical manifestations (with only response rate of 46%) reported recurring problems at the paranasal sinuses in 59% of all patients and 69% of these patients needed corresponding surgical intervention [17]. A systematic nasal endoscopic examination, as performed in our ENT clinic, provides a more objective approach. A typical nasal sign in endoscopy is the presence of sticky secretions strung between the septum and the inferior turbinate. Nasal polyps in PCD have already been reported in the literature as occurring in 25% to 52% of adults [11,18] which is similar to our findings. The prevalence of nasal polyps in PCD is similar to cystic fibrosis (prevalence of 39-48%) [19,20]. Nasal polyps in PCD differ from the classic eosinophilic sinonasal polyposis, since they are associated with oedema, congestive mucosa and sticky or purulent secretions. Such an atypical subtype of rhinosinusitis is mainly found in rare conditions such as cystic fibrosis and immunodeficiency which represent the classical differential diagnosis of PCD [21,22].

Sinus CT scan analysis showed sinus hypo- and/or aplasia in around a third of our patients,(particularly frontal and sphenoid sinuses),which is lower than in the series of Pifferi et al. and El Sayed et al. (73% and 56% of abnormalities, respectively) [23,24]The sinus CT scan can be normal in adults with PCD as described in our se. As previously reported, we found sinus opacities in almost all of our patients [25]but a normal sinus CT scan can be normal in adults with PCD as described in our series.: more specifically, almost all of them exhibited generalized sinus opacities mainly involving the maxillary and ethmoidal sinuses.However, sinus CT opacities were mainly incomplete.Consequently, as already described [16], the mean CT-score was lowlow, even ),which contrasts with the reported score in non-PCD patients with chronic sinusitis and asthmaa (mean score of 19.5) [26]. A high incidence of lung colonization by Pseudomonas aeruginosa has been reported [28-30].The concordance between sinus and lung bacteria in PCD, as in cystic fibrosis, raises the hypothesis ofsinuses functioningas a bacterial reservoir leading to chronic lung infection [31,32].The most striking argument sustaining this hypothesis is lung re-colonization from the sinuses which is frequently observed in lung-transplanted cystic fibrosis and PCD patients [33]. Moreover, sinus surgery seems to improve lung colonisation [31,33].

Ear disease appears to be less severe in adults with PCD than in children. OME, which is present in nearly all children with PCD [4], was found in around a quarter of our adult patients. This contrasts with retrospective study of Anderson et Al. [34] who found 68% of ears with OME in the age group > 17 years (n=33) but the mean (interval) age of the group was not given and could introduce a bias in comparison with our adult patients [34]. Ear disease would thus appear to be less frequent than sinonasal disease in adults, confirming the significant improvement of ear disease with age[4]. Unlike children, very few adults complain of otorrhoea and earache in our series ins. Thiscontrastswith the study of Anderson et al. [34]who found 32% of patients having chronic otorrhoea but only if they had ventilation tube insertions. This difference could be explainedby aehigher percentage of patients with ventilation tubes in series of Anderson et al. than in our series (54% versus 37.5% respectively).could explain this difference.In our adult series, the main otologic symptom washearing losswhich measuredwith audiometrywas present in half oftheour patients, in the same line witha previousthe questionnaire-based studyy of 44 adult patients with PCD, which reporteda permanent auditory impairment in 58% of the participants[17]. Anderson et al. [34],  noted hearing significantly improving with increasing age, but looking their results into details show that only. However, in this study, only air bone gap andair conduction improved with increasing age whilewhilebone conduction worsened, suggesting that only the conductive part of hearing loss improved [34]as found in the present study. Indeed, in our series, we found that the subtype of hearing loss is different in adults compared to children: adults tend to have sensorineural hearing loss (as in one third of the patients in our study) while children with PCD mainly have conductive hearing loss secondary to OME. The pathogenesis of sensorineural hearing loss in the adults is not clear. Presbyacusis was only observed in one patient who was 60 years old. Ototoxic adverse effects of routine PCD therapies such as aminoside or macrolides could be involved or the passage of bacterial toxins into the inner ear [35]. Lundman et al. demonstrated in chinchillas that Pseudomonas aeruginosa exotoxin A can cross the round window, causing irreversible hair cell loss [36]. Moreover, despite chronic otitis in childhoodand tympanic retraction, no complications such as cholesteatoma were reported in our PCD cohort in line with other studies (only one case cholesteatoma in a PCD patient described ias in the literature) probably because of a different physiopathology[24]. Interestingly, a significant correlation was found between the presence of OME and lung function in our study. This result needs to be confirmed but it could suggest that OME in an adult with PCD could be a severity marker of lung function and lead to closer follow up and more aggressive treatment.

Compared to lower airway disease in PCD, the natural course of upper airway disease is less known in adults.  Nasal symptoms were more severein≥ 40-year-old patients in our study suggestingWe observed a significant worsening of sinusitis with age. For most patients, nasal surgery was performed early in the natural history of ENT disease, and mainly before PCD diagnosis. In a recent adult study, 52% of patients had sinus surgery at least once [18]. Once PCD diagnosis was established, the type of sinonasal surgery focused on the inferior turbinates and ethmoid to improve nasal obstruction and rhinorrhoea [37]. It is important to note that sinonasal surgery could complicate nasal sampling for ciliary analysis and thus delay PCD diagnosis. After PCD diagnosis, very few patients underwent ENT surgery for chronic rhinosinusitis and none for ear disease.

This study highlights that ENT data could be usedsed in adults to propose a newspecific adultclinical score to select patients for PCD diagnostic testing [38]. Indeed, the clinical score used in pediatric populations is probably less appropriate in adults. Moreover, since disease presentation changes andretrospective clinical informationcollected in adults are less relevant than those collected in children, especially concerning neonatal respiratory distressre lacking. For example, we found that OME, which is a hallmark presentation of PCD in children, was uncommon in the adult. Along the same lines, we found an incidenceof neonatal respiratory distress < 30% which is less compared to the paediatric studies that estimate an incidence of 44 to 85%[39,40].”

 Minor Revisions

 POINT 8.     Study type: Study design – In the abstract it is stated this is a retrospective study (line 36), however in the ‘Experimental Section’ line 71 states this is a “retrospective observational study.”  By definition an observational study must be prospective. For this study the word “observational” should be deleted.

Response 8: We agree with your pertinent comment and we changed the sentence  line 75. The word “observational” is now deleted.

« We performed a retrospective study in PCD patients »

POINT 9.     If used, a copy of the standardized collection form should included in the supplement.

Response 9: For this retrospective study, only a database was extracted from the patient’s standardized files without a collection form.

POINT 10.  Text language and typographical errors:

Line 62. “unspecific”, should be  “non-specific”

Line 93. FEV1 = forced expiratory volume’ to state ‘forced expiratory volume after one second”

Line 272. “aminoside” should be aminoglycoside.

Line 277 insert “of” one case of cholesteatoma

Response 10: we thank you for your linguistic revision. We modified all sentences except the last sentence already deleted in order to summarize the discussion

Round  2

Reviewer 1 Report

The authors have appropriately responded to the reviewers's comments, and the manuscript has been greatly improved.

Reviewer 2 Report

the authors were very responsive and have improve the manuscipt